# Plants on Rich-Magnesium Dolomite Barrens: A Global Phenomenon

**DOI:** 10.3390/biology10010038

**Published:** 2021-01-08

**Authors:** Juan Mota, Encarna Merlo, Fabián Martínez-Hernández, Antonio J. Mendoza-Fernández, Francisco Javier Pérez-García, Esteban Salmerón-Sánchez

**Affiliations:** 1Departamento de Biología y Geología, CEI·MAR and CECOUAL, Universidad de Almería, 04120 Almería, Spain; jmota@ual.es (J.M.); emerlo@ual.es (E.M.); fmh177@ual.es (F.M.-H.); amf788@ual.es (A.J.M.-F.); fpgarcia@ual.es (F.J.P.-G.); 2Departamento de Botánica, Unidad de Conservación Vegetal, Universidad de Granada, 18071 Granada, Spain

**Keywords:** glade, dolostone, serpentine, gypsum, Mg, endemism, plant ionomic

## Abstract

**Simple Summary:**

Little is known about the relationship between dolomite soils and the flora that develops on them due, among other reasons, to their diffuse separation from limestone, and the lack of a common approach and terminology. Thus, the main aim of the present review was to define what the dolomite phenomenon is, delimiting its global extent, and establishing its relationship with other edaphic phenomena on serpentine and gypsum. To achieve it, in addition to the information compiled by the authors in previous research, an extensive review relative to this topic was performed. This study’s results showed that the “dolomite phenomenon” occurs globally and is evidenced through the appearance of an endemic flora on nutrient-poor soils with high levels of magnesium. Dolomite habitats cause adaptations in plants to be more recognisable than others occurring on more bizarre rocks. Unfortunately, they have been poorly studied from an ecological, evolutionary and conservational point of view. Therefore, the definition of its universal demarcation and characteristics becomes necessary. The present review is a starting point to reach this goal.

**Abstract:**

For botanists and ecologists, the close link between some plants and substrates, such as serpentine or gypsum, is well known. However, the relationship between dolomite and its flora has been much less studied, due to various causes. Its diffuse separation from limestone and the use of a vague approach and terminology that, until now, no one has tried to harmonize are among these reasons. After carrying out an extensive review, completed with data on the distribution of plants linked to dolomite, the territories in which this type of flora appears at a global level were mapped using a geographic information system software. In addition, data on soils were collected, as well as on their influence on the ionomic profile of the flora. These data were completed with the authors’ own information from previous research, which also served to assess these communities’ degree of conservation and the genetic diversity of some of their characteristic species. The results showed that the so-called “dolomite phenomenon” is widely represented and is clearly manifested in the appearance of a peculiar flora, very rich in endemisms, on dry soils, poor in nutrients, and with a high Mg level. Although dolomite habitats cause adaptations in plants which are even more recognizable than those of other rock types, they have not been widely studied from an ecological, evolutionary, and conservation point of view because, so far, neither their characteristics nor their universal demarcation have been precisely defined.

## 1. Plants on Rich-Magnesium Dolomite Barrens: A Global Phenomenon

According to Cooper and Etherington [1], the suggestion that Ca:Mg ratio is an important factor in plant growth was first made by Loew and May [2]. This relationship may help to understand plants narrowly linked to a specific substrate along a wide gradient ranging from ultramafic rocks, in which Mg predominates over the Ca, to gypsum, where the opposite occurs. Limestone and dolomite can be found between these two extremes. Dolomite is a carbonate rock composed of calcium magnesium carbonate, typically CaMg(CO_3_)_2_. The mineral that mainly composes it is also called Dolomite. For that reason, some geologists prefer to name this rock dolostone avoiding confusion between both terms. However, this is not the only disputable issue. The boundaries between what a limestone and a dolostone are can also be diffuse. According to Chilingar and colleagues [3], limestone is a sedimentary rock containing more than 50% of calcite and dolomite, with a dominance of the former; in the case of dolostone, dolomite is predominant. Obviously, there is a gradient between both types of rocks that leads to a discussion about dolomitic limestone and calcitic dolomite. Considering these criteria, the end-member dolomite (without any particular qualification) contains more than 90% of the mineral dolomite.

Dolomite rocks or dolostones are widely distributed over the planet’s surface [4], although they do not always appear clearly differentiated from limestone in available geological cartography. This imprecision does not affect other types of rock such as gypsum or ultramafic rocks, which are much better registered in geological cartography [5], but it does hinder the study of dolomitic flora and vegetation (e.g., [6,7]). Despite this, there are numerous publications that mention plant communities associated with dolomite and dolomitic limestone (e.g., [1]). All of them recognize the influence of this type of rocks both on flora and vegetation, and on the landscape [8]. However, the study of what some researchers have called the “dolomitic phenomenon” [9,10,11] is not an easy task, considering the heterogeneity of habitats included. Indeed, Mota [12] distinguishes up to seven rocky environments with dolomite-associated vegetation. Among all of them, perhaps cliffs are the most universally widespread and homogeneous, because gravity is the prevailing ecological factor, due to its verticality [13]; such environments also comprise several associated microenvironments (overhangs, vaults, and ledges) which harbour different types of plant communities. Debris or stony grounds are frequent at the foot of these cliffs, more or less mobile depending on the slope, caused by mechanical weathering, which give rise to highly specialized plant communities [14]. Although not as markedly as in cliffs, gravity plays a very important role in these habitats, especially because it reduces the availability of water, thus the chemical composition of the rock is not as decisive as in other places [12]. However, so-called sweeps or glades are those habitats in which the link between plants and dolostone is most exacerbated. The difficulty of using these terms, in an international context, lies in that they have been used imprecisely and almost always to refer to vegetation types in North America [8,15]. The “typical” dolomite barrens (an extent of land sparsely vegetated) are well defined by their lithology (dolostone, marly dolomites or dolomitic marbles), tectonics (affected by strike-slip faults), the predominant texture of their soils (fragmented rocks ranging from centimetre to micrometre sizes) and by the frequent effect of strong erosive and meteorization processes (thermal cycles, karstic dissolution) which often leads to ruiniform reliefs [16]. Flora extraordinarily rich in endemic species appears in association with these strongly weathered bedrock outcrops, as in gypsum and serpentine areas, although it has been much less studied.

Paraphrasing Goethe “We see only what we know”, and thus, the objectives of this research are to:Define as precisely as possible what the so-called dolomite phenomenon is (dolomite edaphism, dolomitophily)Delimit its global extentEstablish its relationship with other edaphic phenomena in which Ca:Mg ratio is decisive with the aid of:3aThe edaphic characterization of the substrates in which it occurs and3bThe ionomic composition of plants that grow on them and the known mechanisms that explain homeostasis and the efficiency of Mg’s use in plants (as a major element in dolomite)Raise issues related to the conservation and genetic diversity of this type of flora, very rich in endemisms, many of which are local and threatened.

## 2. Definition of the Dolomite Phenomenon

Despite their geographical amplitude and geomorphological and climatic heterogeneity, there are a number of features that are common to all descriptions of the vegetation associated with dolomite. These traits can assist in delimiting the dolomite phenomenon (dolomitophily or dolomite edaphism, sensu Mota, and colleagues [17,18]).
There are patches of exposed dolomite (or dolomitic marble or dolomitic limestone) bedrock, associated with thin and undeveloped soils, on which they become frequently disaggregated rock fragments providing a gravelly or even sandy appearance to their surface [19,20,21]. The pebble or even sandy appearance of these soils results from the fact that they occur in heavily tectonized areas [17]. This geological process is associated with another climate process that also contributes to generating such debris by the mechanical breakdown of the rock, promoting brecciation, disintegration, and the formation of dolomitic sands. Frost shattering [22] and the thermal expansion of these rocks at high temperatures [23] appear to be the dominant local weathering processes. These features lead to strong edaphical stress and prevent the surrounding vegetation, usually conifer forests, from succession and closure.As dolomite rocks are relatively slowly weathered, these soils are usually shallower, can thus hold less water and by way of consequence, have a lower capacity for nutrient supply. This feature is accentuated in south-facing, and frequently steep slopes and ridges which, together with the textural characteristics of the soil (from pebbly silt loam to coarse rubble) and high insolation, promotes erosion and drainage. At least in areas with a Mediterranean climate, summer soil moisture levels are extremely low. In general, glades are drought-prone, which offers conditions hostile to not adapted plants; consequently, they represent sharp and obvious discontinuities with the nearby vegetation [24].Dolomite soils show a soil exchange complex which is dominated by Ca and Mg, but they differ chemically from their non-carbonate counterparts primarily in that they have a higher pH, and lower Fe, P and K. Moreover, these soils are unlike limestone-derived ones in their highest proportion of Mg [10,17]. In general, these are nutrient-poor soils with low water retention capacity, which makes these communities unproductive in relation to the surrounding vegetation.Such a habitat calls for specialized adaptations, promoting endemism [25,26]. In these microclimate-soil areas, there are species which are very rare or absent in other places and, in many cases, have a marked relic character, likely due to a lack of severe competition. This is because they disproportionally contribute to regional plant diversity [27], especially in biodiversity hotspots [17,28,29,30].This type of communities is, almost always, easily identifiable due to the physiognomic features and the adaptations shown by the plants composing them. Such adaptations are a consequence of an adaptive convergence process. In most cases, these are open dwarf communities dominated by tough perennial herbs which form flat mats and cushions, frequently silvery white-haired. Some authors have highlighted their convergent adaptive appearance with the dune vegetation [31,32].

## 3. The Extent of the Dolomite Phenomenon

According to the references found, edaphism on dolomite is widespread, as is the case with that related to gypsum and serpentine. In the USA, several authors have used terms such as barren, glades, limestone prairies, and xeric limestone prairies (XLPs) to refer to different types of open communities associated with exposed bedrock among which dolomites are frequent (e.g., [24]). Baskin and colleagues [15] attempted to resolve inconsistencies in the use of such terms by restricting them to those substrates developed on calcareous bedrock and adding that many of them are extremely high in magnesium. In addition, so as to distinguish these barrens from those derived from serpentine and diabase, with comparable or higher Mg levels, they mention the alkaline pH of the dolomites in contrast to that acidic of serpentines.

Curiously, American barrens and glades have not been clearly related to one of the most iconic forests in North America, the bristlecone pine forests, developed mostly on dolomite [8]. The reason for this may be that these are forests in the Alpine region, subjected to very different climatic conditions from other types of barrens and glades. However, that relationship is undeniable, as Billings [33] recognized when pointing out the “desert-like” dolomite barrens as the sharp distinction in vegetation and flora of the White Mountains. Plant communities linked to dolomites have also been mentioned in Central Europe. This relationship was baptized as the “dolomite phenomenon”, including the vegetation types which were formed under the influence of dolomite [9,11,34]. To the south, in the circum-Mediterranean area, especially in middle and high mountain areas, this geobotanical phenomenon is clearly visible from the Baetic ranges to the Taurus Mountains [35,36], with extraordinary representations in the Balkans [37,38,39], Crete [40] and also in the Rif and the Middle Atlas [21,41]. The Alps, the Apennines and the Madonia (Sicily) show many endemisms restricted to this type of habitat [42]. Interestingly, this phenomenon has not been expressly mentioned in the Dolomitic Alps [43], perhaps since it is not as accentuated as in the rest of the mountains mentioned, of a more xeric and Mediterranean nature. In all these territories, the areas richest in endemic plants, associated with dolomite, coincide with those of great tectonic activity. In the Baetic ranges (Spain), the coincidence that exists between the dolomitic outcrops richest in endemisms, many of them local, and the distribution of the Tortonian and Quaternary faults [44] is surprising; the same occurs throughout the Mediterranean basin [45].

In the southern hemisphere, the dolomite phenomenon or dolomitophily (according to Mota and colleagues [17]), has been noted in South Africa [7,26,46] and, to a lesser extent, in Australia and Tasmania [31]. There are also allusions to South America, although not very precise [47]. In Asia there are no excessively specific references, but it seems to occur in some mountains in eastern Anatolia, in the Irano-turanian region [48], and also in Tibet and Himalaya [49,50,51]. However, in the latter territories it is not easy to separate the presence of dolomite and dolomite marbles from that of various types of ultramafic igneous rocks. In the eastern part of Asia, vegetation associated with different types of karst related to limestone and dolomite is mentioned, although under a tropical climate [6]. This circumstance highlights the importance of not only lithology, but also geomorphology and weathering, to explain the relationship between plants and magnesium rocks [17,34].

Therefore, climate is another element to take into account when considering edaphism on dolomite, and not merely in terms of rainfall. This geobotanical phenomenon has been alluded to in the White Mountains of California, in the Alps, and in several Mediterranean mountains. All these territories are characterized by their very cold winters, a trait they share with the so-called “alvars”. Alvars are globally uncommon ecosystems distinctive for their unusual plant species’ composition and natural openness (open scraped alvar) [52]. They are present on thin or nearly absent soils underlain by flat limestone or dolomitic bedrock [52]. They are documented in Scandinavia, the northern parts of the United States, and Canada [53]. Despite the northern latitude of these ecosystems, they contain a good number of threatened and endemic species [54]. Due to their special features, bedrock that restricts drainage, they are subject to extreme variations in moisture availability that range from drought conditions to periodic flooding [52]. Alvars are widely distributed, including areas of Greenland [55].

Definitely, although the data for dolomite edaphism are much more imprecise than for serpentine or gypsum, there is no doubt that it is a global phenomenon (see above references) when comparing the documented distribution of these three geobotanical phenomena (Figure 1). 

## 4. Dolomitophily and Other Ca:Mg Edaphisms

### 4.1. Dolomitic Soils

Dolomite glade soils are characterized by being shallow and poorly developed overlying massive bedrock, a higher albedo, sandy texture, poor water retention capacity, alkaline pH (almost always higher than 8), and low MO, N, and P contents [64,65,66,67,68,69]. High pH values and low soil moisture contents are the main environmental factors which impair nutrient mobility in calcareous soils [70]. Phosphorus (P) was considered to be one of the key factors for plant productivity on calcareous and gypsum soils, acting as a co-limiting element together with N [6,71,72]. In fact, phosphate-treated dolomite soils significantly increased their productivity with respect to untreated dolomite soils [64].

In dolomitic outcrops from S of Spain, Carreira and colleagues [66] found that P availability for plants markedly changes along with soil and ecosystem development, from open scrub communities to forests. In the former, the geochemical system in which carbonate is dominant, P is immobilized by Ca; yet, along the vegetal succession, it develops into a system controlled by Fe and Al oxides, coupled with the recycling of P through organic matter. According to Campillo and colleagues [67], these changes associated with plant succession affect not only P, but also MO, K, and N, in addition to water retention capacity. Regarding the latter parameter, recent research has revealed that carbonate rocks store measurable amounts of water compatible with extraction and use by plants’ roots. Although available water content (AWC) is very low, (around 5 mg g^−1^ per dolostone), it might be sufficient to maintain plant hydration under prolonged drought [73,74]. 

Table 1 allows for the comparison of the aforementioned parameters, as well as others, which serve to define this type the dolomite soils. Among these, both their high percentage in carbonates, which is frequently above 60%, and their exchange complex dominated by Ca and Mg stand out (see references above). The proportions of Ca:Mg in dolomite are intermediate between serpentine and gypsum, and very close to those of limestone. As far as pH is concerned, among all the soil types mentioned before, the highest values (8.3) were found for dolomite and the lowest (6.8) for ultramafic [17].

### 4.2. Ionomic Aspects

Recent research suggests that knowledge of plant ionome is a key aspect to globally understand the ecosystems of the Earth [77,78]. Much of the research has been focused on this aspect in serpentine plants (e.g., [79,80]) and gypsum (e.g., [72,81]). Attention has been focused on several elements, often considered key, and on their ratios. This is the case of C:N or N:P:K. ratios. For the latter three elements, the diagrams of Olde Venterink and colleagues [82] have helped to understand the limitations that each of these nutrients can pose for plant growth [83,84]. N:P ratio has also been widely used [85], for which an optimal range of 10–20 was indicated. Values above or below these might indicate N or P deficiency, respectively [86]. However, nutrient concentrations and N:P ratios varied widely among the species and sites [85]. Table 2 shows the values of this relationship for several plants that grow on dolomite. In them, dolomitophiles (with a close relationship with dolomite) have been distinguished from non-dolomitophiles [17]. As in Figure 2, P seems to be the most limiting nutritional element in this type of soil. Probably, the affinity of this type of plants for dolomite may be due to the fact that they are more tolerant to these low P levels [66], among other causes. In fact, the values for this element in these plants (0.072% over dry weight) are far below the mean values that are interpreted as normal (0.102% over dry weight [85]; 0.123% DW [77]; 0.195% DW [87]), although they fall within the wide range established by Kattge and colleagues [77]. In the case of N, the values are only slightly below the means and, unlike P, there are no significant differences between dolomitophytes and non-dolomitophytes (Table 2). The triplots in Figure 2 (and Table 2) show that this P limitation also occurs in plants growing on gypsum as the high proportion of S could make it difficult for plants to acquire P [88]. However, the same is not true in serpentine, where NPK values seem to be more balanced (Figure 2); the same can be said for N:P ratio, which falls within the range indicated by Gosewell and Koerselman [85] (Overall means were 13.3 mg g^−1^ for the concentration of N (Nconc), 1.02 mg g^−1^ for Pconc, and 13.3 for N:P ratio; see Table 2), although in this case it should be noted that few data are available in this regard (but see [79,80,89]).

In the case of serpentine, together with the presence of heavy metals, Ca:Mg ratio has been widely used to interpret the so-called serpentine syndrome and to understand the limitations that plants may suffer in these soils [94,95,96]. As already noted, this relationship is considered key to understanding the link between plants and some types of rock. Mota and colleagues [18] have suggested that the relationship between both elements in plant tissues may serve to approximate a geoecological theory or, at least, to explain not only the edaphism of serpentine, but also that of dolomite and gypsum. These edaphisms can be ordered in a series ranging from lowest to highest Ca:Mg ratio, in serpentine (ultramafic rocks), dolostone, limestone, and gypsum; this gradient can also be recognized on the ground (Table 1). In the case of gypsum, Merlo and colleagues [72] have incorporated a new element into the discussion: S, very abundant in this type of soils and not so in those of dolomite and serpentine. Sulphur is now recognized as the fourth major plant nutrient after N, P, and K globally [97]. Ca:Mg:S ratio can help differentiate the geo-ecological niche of plants in these three soil types. This relationship may also be represented in ternary plots, as in the case of N:P:K ratio. In this, the lines that mark the limitations for one or the other element have been drawn according to the values established by Merlo and colleagues [72]. According to this analysis, S seems to be the candidate element to limit the growth of plants in dolomite from the Baetic ranges, although this fact is not so intense when analyzing dolomite and limestone from the Hungarian Middle Mountain-Range [10], probably because of the different climatic conditions, especially as regards its greater amount of precipitation. In the case of gypsum, Mg is the main candidate for a limiting element whose scarcity would have to be tolerated by plants. However, Mg contents of gypsum plants are not particularly low. The increase in Ca concentration, up to a certain level, also increases the absorption rates of Mg (and also K), although if the increase is very high, it may compete with its absorption [98]. In the case of serpentine plants, Ca is the one element that could impose growth restrictions (Figure 3). In this latter case, Mg could replace Ca as an osmotic element to maintain plant growth, as demonstrated in *Arabidopsis thaliana* [99]. Mg requirements for optimal plant growth range from 1.5–3.5 g kg^−1^ in vegetative parts, and between 125 mM and 8.5 mM in soils in order to achieve sufficient supply for plant growth [98]. Fekete and colleagues [10] found values in dolomite plants of 3.14 g kg^−1^, higher than those of limestone (2.56 g kg^−1^). In the case of dolomite plants from the Baetic mountain ranges [68,90], Mg presented average values of 4.6 g kg^−1^ for all species studied, although they were higher for dolomitophile species (7.1 g kg^−1^). Although these data reinforce the physiological and molecular evidence of Mg homeostasis in plant cells [100], they also suggest adaptations such as those proposed by Tyndall and Hull [101] for plants growing on serpentine, such as a higher tolerance to low and high concentrations of Mg, a higher Mg requirement for maximum growth, mechanisms to decrease Mg absorption and Mg exclusion from leaves. The compartmentalization of ions, including Mg, in cellular organelles and in the apoplast of the cells of the tissues of the root and leaves, mainly in the vacuoles, should be added to these mechanisms. The excess may be stored in crystalline phases, which could explain the presence of dolomite (mineral) in *Helichrysum tyrrhenicum* subsp. *tyrrhenicum* [102]. In this species, weddellite (calcium oxalate) has also been found, one of the most common biomineral in higher plants. In the case of gypsophile plants, this type of crystals is common [92,103] and, although its role is still controversial, it may have to do with the excess of Ca on the ground [104]. Ca:Mg ratio is important for plant growth and becomes so narrow that, in some cases, signaling and signal transduction pathways for Ca and Mg response are common [100,105].

The knowledge of the processes in which Mg intervenes may help to understand some of the plants’ adaptation mechanisms to serpentine and dolomite. For example, Mg increases the transport of sugars from the leaves to the roots, which favors the growth of the latter, modifying the root/stem ratio and increasing the water absorption surface [106,107,108]. In fact, many dolomitophile form silver plated buns and have small felted leaves in a way that decreases the transpiration surface [17]. The high Mg content in these habitats may imply a greater resistance of plants to high temperatures and light intensities, very frequent in those environments where they appear within this type of habitat [98]. Since temperatures that can exceed 50 °C [109] have been measured in these types of soils, whereas plant leaves hardly reach 40 °C, transpiration cooling could be the most important physiological mechanism for the adjustment of leaf temperatures below the critical temperature limit in order to avoid heat damage [110]. The most widespread adaptive syndrome in this type of plant, the dense white sericeous indumenta covering its leaves, seems to point in the same direction due to its reflective function.

## 5. Conservation and Genetic Diversity

There are many reasons to rank many edaphic conditioned communities as priority habitats in the Directive 92/43/EEC, as is the case of the flora and vegetation associated with gypsum [111,112]. However, the same attention has not been paid to dolomitic shrublands. In the comparison made by Medina–Cazorla and colleagues [28] between the Iberian habitats of gypsum and dolomite, regarding the relevance of their conservation, authors highlight many similarities and some differences. Their peculiar floras rich in endemisms, alkaline carbonated soils, disjunct distribution and the number of their endangered species, range among the former. However, some of these characteristics are accentuated in dolomite. While in the case of gypsum, between 30–40 species can be considered exclusives to this type of soil, this figure rises to 70–80 for dolomite; nine of the species related to gypsum shrublands are included in the Spanish Red List, while there are only 39 in the case of dolomite [28]. These figures probably have everything to do with the fact that the number of local endemisms is much higher on dolomite. This high richness in endemic species also occurs in other parts of the Mediterranean basin [30,113,114] and the world [25,26], areas in which protection measures for this type of habitat have been claimed [26,42,54,113].

The disjunct distribution of dolomite outcrops conditions the island-like distribution of dolomitophile species, which can be said to respond to a “sky islands” model [115]. In this type of habitat, geographical isolation is the factor that most limits the genetic flow in this type of habitat, creating opportunities for both genetic and phenotypic divergence among populations through genetic drift [116]. As a consequence of this isolation, plants would be expected to show low levels of genetic diversity at the species level [117]. On the other hand, environmental heterogeneity between the “islands” can intensify these processes of divergence [118]. These barriers among fragmented habitat areas, both ecological and geological, frequently induce genetic differentiation [119], which would help explain the existence of high endemicity rates [120]. It is also important to consider the characteristics of dolomitophile flora. Although they may have a small population size, the effective size of populations may be larger than expected, as most dolomitophytes are perennial [90]. Moreover, these plants prefer outcrossing to selfing, at least in Baetic ranges [121]. This is what the limited data available on dolomitophile species seem to indicate. It is generally accepted that rare species tend to show low levels of genetic variability due to their small population size [122], although the existence of high genetic diversity is also possible [123]. Thus, for example, average genetic diversity levels within the population (see Appendix A) are similar to those found in endemic and gypsophile species [92]. In contrast to the aforementioned species, the dolomitophile *Viola cazorlensis* [124], *Helianthemum apenninum* subsp. *estevei* and *H. panossum* [125] show higher heterozygosity values. According to the authors, this may have resulted from recent changes in genetic flow and population size. These changes could be consequence of a recent reduction in population size together with insufficient time for isolation, or extensive, recurrent gene flow [126], which could be associated with climatic fluctuations during Pleistocene. 

On the other hand, the high values of population differentiation indices found in *Convolvulus boissieri* (F_ST_: 0.395, [127]) and *Jurinea pinnata* (F_ST_: 0.374, [68]) are similar to those found in other edaphic endemic plants, such as gypsophytes [92] and other species found in “island-like” habitats [128], which could be indicative of the low gene flow between populations. This is not surprising given the ecological and spatial profile of the species’ habitat. Instead, in the case of the other species considered, such as *Viola cazorlensis*, this value is significantly low (F_ST_: 0.134, [124]). In this species, populations might have fragmented recently, and did not have enough time to differentiate [124]. Moreover, the perennial nature of this species may have played a part in this low differentiation rate. With respect to Californian species, they also show low interpopulation differentiation values (G_ST_; 0.12 in *Erigeron parishii*; 0.01 and 0.07 in *Astragalus albens* and *Eriogonium ovalifolium* var. *vineum*, respectively). Low values could be explained because of the restricted distribution of the species, and the low distance among their populations [129,130,131].

## 6. Conclusions

The so-called dolomite phenomenon, also known as dolomite edaphism or dolomitophily, is widespread throughout the globe. It occurs when dolomitic rocks emerge, especially with high Mg content, and when tectonic or weathering processes generate skeletal soils, predominantly sandy or gravelly in texture, which further complicates water retention and as a result, makes them very dry. Although these types of habitats can vary greatly depending on the type of climate under which they develop, they are associated with a specific flora, also accompanied by other tolerant species. In the case of Mediterranean high mountain areas, where this phenomenon reaches its most striking expression, dwarf compact scrubs with small felted leaves predominate, disposed as a silvery mosaic of hairy rugs. In other territories the reduction in the size of the plants, the shortening of their stems in favour of root development and the presence of miniaturized leaves is also evident. Among their notable adaptations to inhabit such harsh environments, their tolerance to Mg and to the low content of soil nutrients, especially P, in addition to their resistance to hydric and thermal stresses, should be noted. The study of the ecophysiological mechanisms that underlie these adaptations could help to better understand Mg metabolism, especially in alkaline soils, very different from those on ultramafic rocks. Due to their richness in endemisms, many of them local, this type of habitat deserves greater attention from the point of view of conservation and in all areas, from the genetic to the community perspective.

## Figures and Tables

**Figure 1 biology-10-00038-f001:**
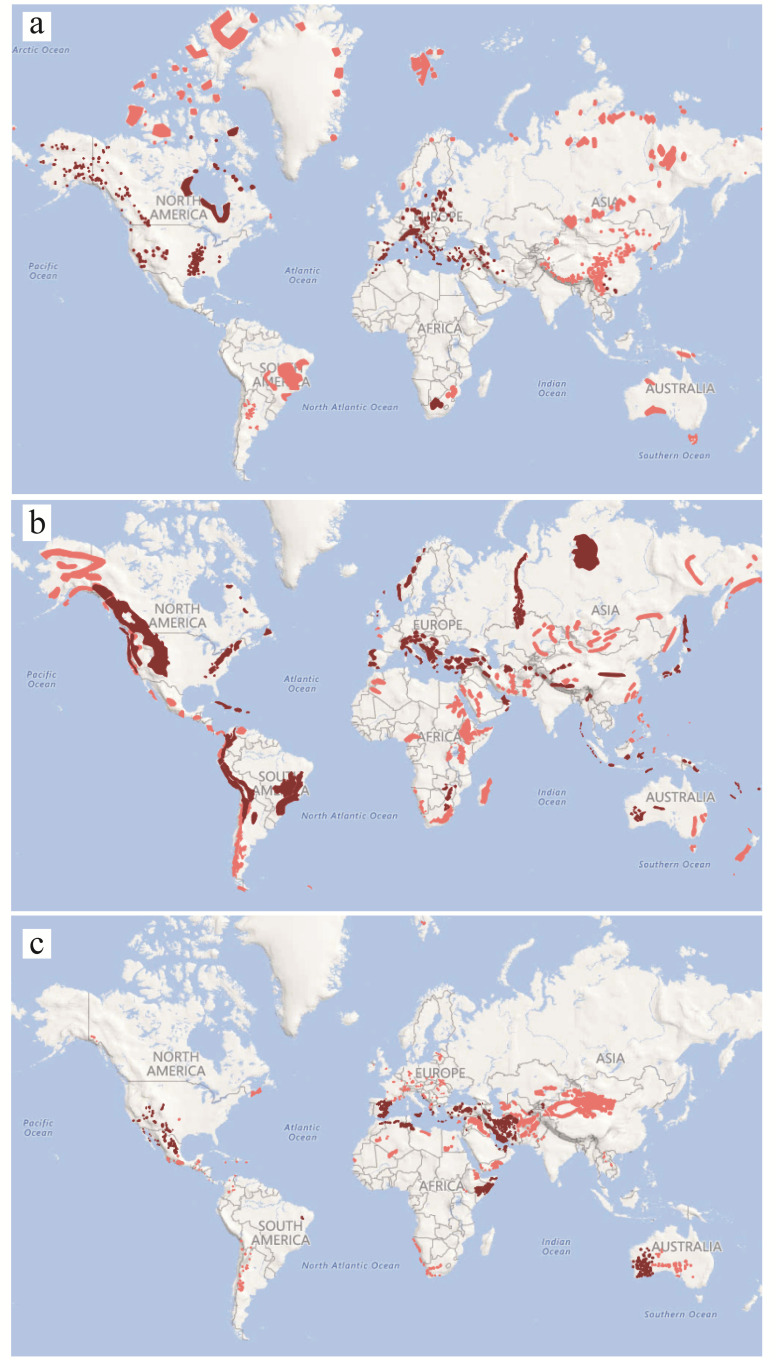
Worldwide distribution of: (**a**) dolomite outcrops; in dark those in which there is associated dolomiticolous flora (based on [56,57,58] and other references in the main text); (**b**) outcrops of ultramafic rocks; in dark those in which hyperaccumulator plants have been detected, especially Ni (based on [59,60,61]); (**c**) gypsum outcrops; in dark those in which associated gypsophile flora has been documented [62,63].

**Figure 2 biology-10-00038-f002:**
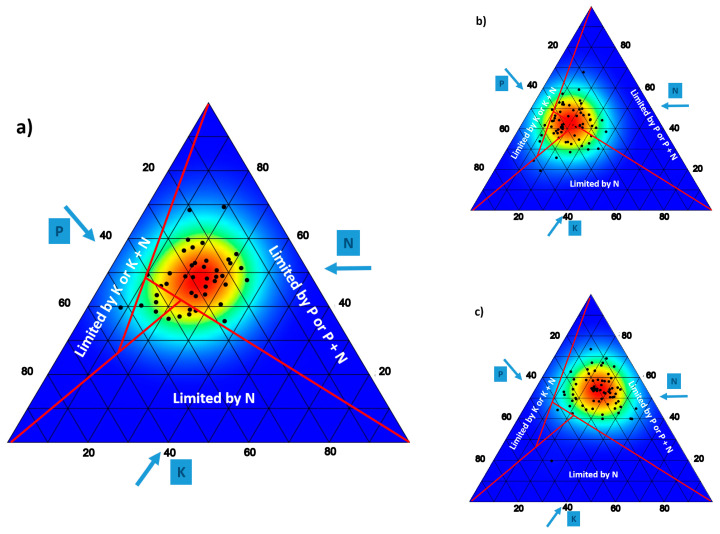
Ternary plot showing the stoichiometric relationship of N, P, and K foliar contents in plants living on: (**a**) dolomite (data from [68,90]); (**b**) serpentine [89]; (**c**) gypsum [92,93]. Lines in the graphs indicate the critical ratios of N:P (14.5), N:K (2.1), and K:P (3.4), and are based in [82]. These lines divide the plot into four sections, three of which indicate N limitation (Limited by N), P or P + N co-limitation (Limited by P or P + N) and K or K + N co-limitation (Limited by K or K + N). For the central triangle section, the N:P:K stoichiometric ratios cannot be used to determine the type of nutrient limitation or this is non-NPK limitation. For visual reasons P concentration is multiplied by a factor of 10. Arrows indicate the direction in which the axes should be read.

**Figure 3 biology-10-00038-f003:**
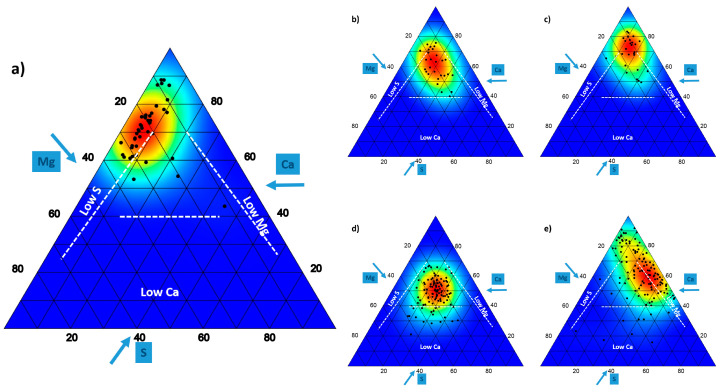
Ternary plot showing the stoichiometric relationship of Ca, Mg and S foliar contents in plants living on: (**a**) dolomite data from [68,90]; (**b**) dolomite [10]; (**c**) limestone [10]; (**d**) serpentine [89]; (**e**) gypsum [92,93]. Dashed lines in the graphs indicate the levels of deficiency for each the elements studied and are based in Merlo and colleagues [72] thresholds. For visual reasons P concentration is multiplied by a factor of 10. Arrows indicate the direction in which the axes should be read.

**Table 1 biology-10-00038-t001:** Textural and chemical parameters mean values (or ranges) of dolomitic soils and others (ultramafic, limestone, and gypsum) with which they have been compared according to the references used here. Gravel, sand, silt, clay, carbonates, organic carbon (OC), N, Ca, and Mg in %; cation-exchange capacity (CEC), Ca^2+^, Mg^2+^, Na^+^, and K^+^ in cmolc kg^−1^; WR = water retention capacity. * includes silt + clay.

	n	Gravel	Sand	Silt	Clay	pH	Carbonates	OC %	N (%)	Ca (%)	Mg (%)	Ca:Mg	CEC	EC	Ca^2+^	Mg^2+^	Na^+^	K^+^	WR
Dolomite																			
[25]						7.5				6.89	0.999								
[11]	5					7.68	67.87			16.2	7.3	2.22							
[67]	4	42.61	71.9	21.00	7.53	8.6	85.43	1.22	0.07						12.63	5.75	0.05	0.11	9.65
[65,66]	4	49.5–86.5	85.8–92.6		1.5–3.8	7.5–8.5	69.4			18.6	10.5	1.77	1.53–10.6						
[64]	4					7.5–8.2	25.4	2.23							32.5	6.22	0.29	0.59	8.2
[75]	1		63.7	34.1	2.2	8.0–8.1							14.3		11.5	3.4	0.1	0.4	
[7]	5					7.6				1.18	0.5296	2.2	20.2						
[76]	2					8.2			0.1	21.9	11.9	1.84							2.7
[68]	14	44.43	51.21		48.79 *	8.2	77.89	3.13	0.26	16.17	10.05	1.61	13.21	1.16			0.04	0.2	6.95
[17]	15					7.94				10.92	5.98	2.56							
Ultramafic																			
[25]	6					6.5				0.918	0.999	0.92							
[17]	10					6.81				8.21	12.32	0.86							
Gypsum																			
[68]	10	14.65	17.77		82.23 *	8.2	31.64	0.54	0.07	17.52	1.37	37.69	9.02	2.59			0.13	0.23	15.47
Limestone																			
[25]	3					7.3				0.77	0.02								
[11]	15					7.53	54.24			25.74	0.31	95.61							
[17]	14					7.73				24.28	2.89	11.3							

**Table 2 biology-10-00038-t002:** Mean values for N, P, K, N:P, Ca, Mg, S, and Ca:Mg (standard deviation in parentheses) for plants growing in the dolomites of the Baetic mountain ranges [90,91] and Hungary [10], serpentine [79,89], and gypsum [18,72,92,93].

		**n**	**N**	**P**	**K**	**N** **:** **P**
Dolomite_Baetic	Non-Dolomitophytes	142	1.71 (0.82)	0.12 (0.07)	0.82 (0.53)	15.76 (7.14)
	Dolomitophytes	90	1.69 (0.45)	0.07 (0.02)	0.82 (0.19)	25.43 (10.83)
	All	232	1.70 (0.72)	0.11 (0.06)	0.82 (0.45)	18.91 (9.65)
Serpentine		67	1.59 (0.9)	0.14 (0.09)	0.82 (0.07)	12.08 (4.84)
Gypsum		73	2.34 (0.91)	0.09 (0.05)	1.23 (0.69)	30.30 (13.96)
		**n**	**Ca**	**Mg**	**S**	**Ca:Mg**
Dolomite_Baetic	Non-Dolomitophytes	142	2.24 (1.94)	0.46 (0.26)	0.24 (0.32)	4.76 (3.13)
	Dolomitophytes	90	1.98 (0.72)	0.71 (0.37)	0.19 (0.20)	2.95 (2.95)
	All	232	2.15 (1.65)	0.54 (0.32)	0.22 (0.29)	4.17 (2.73)
Dolomite_Hungary		28	1.06 (0.44)	0.31 (0.08)	0.32 (0.16)	3.35 (1.19)
Limestone_Hungary		27	1.39 (0.61)	0.26 (0.12)	0.31 (0.17)	6.00 (3.12)
Serpentine		109	0.43 (0.11)	0.24 (0.12)	0.22 (0.07)	2.16 (1.03)
Gypsum		123	4.87 (3.13)	0.80 (0.77)	2.69 (2.43)	12.36 (20.08)

## Data Availability

Data is contained within the article or Appendix A.

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
