# Peer review of "Plants on Rich-Magnesium Dolomite Barrens: A Global Phenomenon"

_biology, 2021, doi:10.3390/biology10010038_

Round 1

Reviewer 1 Report

The manuscript by Juan Mota et al., entitled “Plants on rich-magnesium dolomite barrens: a global phenomenon”, reports on the current knowledge on the so –called “dolomite phenomenon” trying to precisely define it, delimiting its global extent, establishing the relationships between edaphic features of substrates and the ionomic composition of plants living there and discussing the conservation problems of this particular habitat. It is an interesting contribute to better understand the characteristic of this habitat in different territories of the world.

On the whole, the manuscript reads well, the arguments are well presented, the images are of good quality, literature is extensive. For these reasons, I think that the paper deserves to be published in Biology. However, there are some issues that should be addressed by the Authors before the paper can be accepted for publication.

Specific comments

Keywords: the last keyword is “dolomitic flora”, actually through the text very few data on dolomitic flora are reported in the conclusion section (from line 304 to line 307). As keywords are used for indexing, I suggest to delete it or replace with another keyword most closely matching the content of the article (i.e. plant ionomic).

Section 1 “plants on rich-magnesium dolomite barrens: a global phenomenon”: I suggest to add a clear definition of “barrens”. This could be helpful for readers that approach this argument for the first time.

Figure 2: the legend of a figure should help the reader understand the meaning of a figure. In the present form, there is very little explanation and I find the interpretation of the graphs difficult.

Figure 3: see comments of figure 2; in addition, the red dashed lines are hardly visible

Section 5 “Conservation and genetic diversity”: the discussion about the genetic diversity and the occurrence of endemism seems to be a bit hasty; indeed the causes limiting the genetic flows and the low levels of genetic variability are different in different species because linked to some important factors such as self pollination/cross pollination. Furthermore, the richness of endemism is linked not only to substrate conditions but also to the history of flora. I suggest to broaden the discussion.

Author Response

We would like to thank the reviewer for his appreciation and good criticism of our work. Thank you very much also for the comments and corrections that have undoubtedly helped to improve the manuscript.

Responses to each specific comment

Point 1: Keywords: the last keyword is “dolomitic flora”, actually through the text very few data on dolomitic flora are reported in the conclusion section (from line 304 to line 307). As keywords are used for indexing, I suggest to delete it or replace with another keyword most closely matching the content of the article (i.e. plant ionomic).

Response 1: Following the recommendations of the reviewer, we have deleted dolomitic flora and added plant ionomic.

Point 2: Section 1 “plants on rich-magnesium dolomite barrens: a global phenomenon”: I suggest to add a clear definition of “barrens”. This could be helpful for readers that approach this argument for the first time. 

Response 2: A definition of barren has been included: extend of land sparsely vegetated.

Point 3: Figure 2: the legend of a figure should help the reader understand the meaning of a figure. In the present form, there is very little explanation and I find the interpretation of the graphs difficult. the legend of a figure should help the reader understand the meaning of a figure. In the present form, there is very little explanation and I find the interpretation of the graphs difficult.

Response 3: Figure 2 caption has been rewritten as follow:

Figure 2. Ternary plot showing the stoichiometric relationship of N, P and K foliar contents in plants living on: (a) dolomite (data from [68,90]); (b) serpentine [89]; (c) gypsum [92][93]. Lines in the graphs indicate the critical ratios of N:P, N:K and K:P, and are based in [82]. These lines divide the plot into four sections, three of which indicate N limitation (Limited by N), P or P+N co-limitation (Limited by P or P+N) and K or K+N co-limitation (Limited by K or K+N). For the central triangle section, the N:P:K stoichiometric ratios cannot be used to determine the type of nutrient limitation or this is non-NPK limitation. For visual reasons P concentration is multiplied by a factor of 10. Arrows indicate the direction in which the axes should be read.

Point 4: Figure 3: see comments of figure 2; in addition, the red dashed lines are hardly visible

Response 4: Figure caption has been rewritten as follow:

Figure 3. Ternary plot showing the stoichiometric relationship of Ca, Mg and S foliar contents in plants living on: (a) dolomite (data from [68,90]; (b) dolomite [10]; (c) limestone [10]; (d) serpentine [89]; (e) gypsum [92,93].  Dashed lines in the graphs indicate the levels of deficiency for each the elements studied and are based in Merlo and colleagues [72] thresholds. For visual reasons P concentration is multiplied by a factor of 10. Arrows indicate the direction in which the axes should be read.

In addition, for a better visualization, dashed lines are now white 

Point 5: Section 5 “Conservation and genetic diversity”: the discussion about the genetic diversity and the occurrence of endemism seems to be a bit hasty; indeed the causes limiting the genetic flows and the low levels of genetic variability are different in different species because linked to some important factors such as self pollination/cross pollination. Furthermore, the richness of endemism is linked not only to substrate conditions but also to the history of flora. I suggest to broaden the discussion.

Response 5: The last section of the paper has been improved, discussing about he different causes limiting the genetic flows, specifically in species that grow on dolomitic substrates. Moreover, english style writing has been checked by a native english speaker reviewer. Reviewer will be able to check it in the word document with tracked changes (see attached file).

Reviewer 2 Report

I had the pleasure of reading and revising the Review paper entitled “Plants on rich-magnesium dolomite barrens: a global phenomenon”. This manuscript addresses and explains different topics related to the so-called “dolomite phenomenon”, highlighting the peculiarity of dolomitic environments and the consequent floristic irreplaceability, thus suggesting more attention for their conservation. The text is clear and well written. Please, see the attached document with some minor concerns.

About the supplementary file, this needs improvements:

- Please, explain in the caption that references in numbers are about the ones cited in the main text or just cite them again here and homogenize

-please, align the reference [92], not clear if referring to the gypsum plants

- why the same gypsum plants have lines above/below?

- I suggest using SSR instead of microsatellite, then explicit all the acronyms in the caption

Author Response

Responses to reviewer 2

We would like to thank the reviewer for his appreciation and good criticism of our work. Thank you very much also for the comments and corrections that have undoubtedly helped to improve the manuscript.

Point 1: About the supplementary file, this needs improvements:

- Please, explain in the caption that references in numbers are about the ones cited in the main text or just cite them again here and homogenize

-please, align the reference [92], not clear if referring to the gypsum plants

- why the same gypsum plants have lines above/below?

- I suggest using SSR instead of microsatellite, then explicit all the acronyms in the caption

Response 1: All the suggestions made by the reviewer have been followed. Regarding the references that were not included in the manuscript, now are cited. As a consequence, now, all the references are formatted following MDPI house style.

Point 2: Line 18. at this stage, please explicit Mg

Response 2: Corrected, Mg has been rewritten as Magnesium

Point 3: Lines 27-28  unclear, just mapped? If maintained, the acronym GIS might be explicated

Response 3: Corrected, as follow: “…were mapped using geographic information system software..”

Point 4: line 102 simplify  “…results from these soils occurring in ….”

Response 4: Corrected as follow: “… results from the fact that they occur in…”

Point 5: Section 5. Conservation

-Line 316. Highlighted text: Directive 92/43/EEC, as is the case of the flora and vegetation associated with gypsum [111,112]

-Line 324. Since it is a global-level perspective, please avoid focusing too much on the Spanish case, including other case from other countries/continents

-Lines 327-328. Even if differences are here well exposed, dolomitic communities might be also generically included in 6210, 6220  and under all limestone habitats. Because a limit in the number of habitats is necessary to avoid confusion, I am not sure that incuding a series of new specific dolomitic habitats, might improve their assessment and conservation. I would rather better describe the aready defined habitats. Some thought and possible solutions might be included

Response 5:

We have considered Spanish flora because in this area (and specifically in Baetic Ranges) is where dolomitic flora has been better characterized, and in consequence there area more data available. In addition, following the recommendations of the reviewer, we have internationalized this issue. The following modifications have been made:

We have deleted, following the recommendations of the reviewer, the conflicting phrases related to the degree of protection in Europe (within the framework of the Directive Habitats) of this type of habitat. Moreover, we have internationalized the issue of conservation alluding to other countries and territories around the world. We have rewritten this paragraph as follow: “This high richness in endemic species also occurs in other parts of the Mediterranean basin [30,113,114] and the world [25,26], areas in which protection measures for this type of habitat have been claimed [26,42,54,113].

Point 8: line 342. Explain FST.

Response 8. All this section has been rewritten (see attached word document with tracked changes), including a definition of FST (population differentiation index).

Moreover, English style writing has been checked by a native english speaker reviewer. Reviewer is able to check other minor corrections that were made following the indications of the reviewer, in word document with tracked changes (attached file).
